# Pharmacokinetic/Pharmacodynamic Adequacy of Novel β-Lactam/β-Lactamase Inhibitors against Gram-Negative Bacterial in Critically Ill Patients

**DOI:** 10.3390/antibiotics10080993

**Published:** 2021-08-17

**Authors:** Ruiying Han, Dan Sun, Sihan Li, Jiaojiao Chen, Mengmeng Teng, Bo Yang, Yalin Dong, Taotao Wang

**Affiliations:** Department of Pharmacy, The First Affiliated Hospital of Xi’an Jiaotong University, Xi’an 710061, China; hry15083352653@stu.xjtu.edu.cn (R.H.); sd1341781857@stu.xjtu.edu.cn (D.S.); lisihan@stu.xjtu.edu.cn (S.L.); chenjiaojiao123@stu.xjtu.edu.cn (J.C.); mengmeng123@stu.xjtu.edu.cn (M.T.); Boyang@stu.xjtu.edu.cn (B.Y.)

**Keywords:** novel β-lactam/β-lactamase inhibitors, Gram-negative bacteria infection, critically ill patients, pharmacokinetics/pharmacodynamics, Monte Carlo simulations

## Abstract

The optimal regimens of novel β-lactam/β-lactamase inhibitors (BLBLIs), ceftazidime/avibactam, ceftolozane/tazobactam, and meropenem/vaborbactam, are not well defined in critically ill patients. This study was conducted to identify optimal regimens of BLBLIs in these patients. A Monte Carlo simulation was performed using the published data to calculate the joint probability of target attainment (PTA) and the cumulative fraction of response (CFR). For the target of β-lactam of 100% time with free drug concentration remains above minimal inhibitory concentrations, the PTAs of BLBLIs standard regimens were <90% at a clinical breakpoint for *Enterobacteriaceae* and *Pseudomonas aeruginosa*. For ceftazidime/avibactam, 2000 mg/500 mg/8 h by 4 h infusion achieved >90% CFR for *Escherichia coli*; even for 4000 mg/1000 mg/6 h by continuous infusion, CFR for *Klebsiella pneumoniae* was <90%; the CFRs of 3500 mg/875 mg/6 h by 4 h infusion and 4000 mg/1000 mg/8 h by continuous infusion were appropriate for *Pseudomonas aeruginosa*. For ceftolozane/tazobactam, the CFR of standard regimen was >90% for *Escherichia coli*, however, 2000 mg/1000 mg/6 h by continuous infusion achieved <90% CFRs for *Klebsiella pneumoniae* and *Pseudomonas aeruginosa*. For meropenem/vaborbactam, standard regimen achieved optimal attainments for *Escherichia coli* and *Klebsiella pneumoniae*; 2000 mg/2000 mg/6 h by 5 h infusion, 2500 mg /2500 mg/6 h by 4 h infusion, 3000 mg/3000 mg/6 h by 3 h infusion and 4000 mg/4000 mg/8 h by 5 h infusion achieved >90% CFRs for *Pseudomonas aeruginosa*. The CFRs of three BLBLIs were similar for *Escherichia coli*, but meropenem/vaborbactam were superior for *Klebsiella pneumoniae* and *Pseudomonas aeruginosa*.

## 1. Introduction

Gram-negative bacteria (GNB) are commonly associated with hospital-acquired infections in intensive care units, and *Enterobacteriaceae* and *Pseudomonas aeruginosa* are the leading causes of bacterial infections [1]. However, there has been a dramatic increase in multidrug-resistant (MDR) pathogens among GNB, especially the carbapenem-resistant *Enterobacteriaceae* (CRE), which is a life-threatening infection with mortality rates of about 40% in critically ill patients [2].

Novel β-lactam/β-lactamase inhibitors (BLBLIs) have emerged in this situation. BLBLIs, ceftazidime/avibactam (CA), ceftolozane/tazobactam (CT), and meropenem/vaborbactam (MV) are often administered as salvage therapies for infections due to pathogens that are resistant to most antibacterial agents [3]. Currently, the novel BLBLIs administration is based on their package insert and pharmacokinetics/pharmacodynamics (PK/PD) property. It is well known that the recommended dosage regimen of antibacterial agents is mostly derived from PK studies in healthy volunteers. However, critically ill patients may undergo tremendous pathological and physiological changes [4], which can lead to significant changes in the PK parameters and plasma drug concentrations of β-lactam antibacterial agents [5]. Therefore, the recommended regimen of novel BLBLIs may be contributed to underexposure or overexposure in critically ill patients. Underexposure will increase the risk of drug resistance and mortality, while overexposure will cause adverse events [6,7,8]. Furthermore, some studies have reported that certain GNB showed resistance to novel BLBLIs [9,10,11]. Therefore, the altered PK characteristic and the resistance of pathogenic bacteria make the regimen optimization of novel BLBLIs imperative in critically ill patients.

Monte Carlo simulation (MCS) is now widely used to optimize antibacterial agents’ regimens by combining the pharmacokinetics and pharmacodynamics principles. The results of MCS were expressed as the probability of target attainment (PTA) and the cumulative fraction of response (CFR). PTA was defined as the probability that a specific value of a PK/PD index was achieved at a certain minimal inhibitory concentration (MIC) [12]. For BLBLIs, a joint PTA was calculated, and the joint PTA was the product of the PTA of β-lactam and β-lactamase inhibitors [13]. The CFR was defined as the expected population joint PTA for a specific drug dosage and a specific pathogen [12]. Ceftazidime, ceftolozane, and meropenem are time-dependent antibacterial agents with a PK/PD parameter of the percentage of time during the dosing interval that free drug concentrations remain above minimal inhibitory concentrations (%*f*T > MIC) [14]. β-lactamase inhibitors require a certain threshold concentration (C_T_) to exert their inhibitory effect. The PK/PD parameter of avibactam and tazobactam was the percentage of time during the dosing interval that free drug concentrations remain above threshold concentration (%*f*T > C_T_), and vaborbactam was the 24-h area under the free concentration-time curve divided by the minimum inhibitory concentration ratio (*f*AUC_24_/MIC).

At present, few studies have reported the appropriate regimens of novel BLBLIs in critically ill patients with GNB infections. The objectives of this study for novel BLBLIs were: (1) to evaluate whether the current standard regimens can achieve their PK/PD targets; (2) to identify the optimal regimens in critically ill patients.

## 2. Results

### 2.1. Probability of Target Attainments of Three Novel BLBLIs

Figure 1 showed the joint PTA of three novel BLBLIs in patients who received standard regimens. At a MIC of 8 mg/L, the standard regimen of CA (2000 mg/500 mg/8 h, 2 h infusion) could achieve a ${\mathrm{PTA}}_{\mathrm{CA}}^{50\%}$ (50%*f*T > MIC/50%*f*T > C_T_) of 97.39%. However, the ${\mathrm{PTA}}_{\mathrm{CA}}^{100\%}$ (100%*f*T > MIC/50%*f*T > C_T_) of the standard regimen was >90% only at a MIC ≤ 4 mg/L. The standard regimen of CT (1000 mg/500 mg/8 h, 1 h infusion) yielded a ${\mathrm{PTA}}_{\mathrm{CA}}^{40\%}$ (40%*f*T> MIC/20%*f*T > C_T_) of 92.79% at a MIC of 8 mg/L, and an acceptable ${\mathrm{PTA}}_{\mathrm{CA}}^{100\%}$ (100%*f*T> MIC/20%*f*T > C_T_) was obtained at a MIC ≤ 0.5 mg/L. The ${\mathrm{PTA}}_{\mathrm{MV}}^{45\%}$ (45%*f*T > MIC/*f*AUC_24_/MIC > 9) of the MV standard regimen (2000 mg/200 mg/8 h, 3 h infusion) at a MIC of 8 mg/L was 94.04%, and the ${\mathrm{PTA}}_{\mathrm{MV}}^{100\%}$ (100%*f*T > MIC/*f*AUC_24_/MIC > 9) of the standard regimen was >90% at a lower MIC (≤0.125 mg/L).

### 2.2. Cumulative Fraction of Responses

Table 1, Table 2 and Table 3 showed the CFRs of three novel BLBLIs regimens at different PK/PD indexes for *Enterobacteriaceae* and *Pseudomonas aeruginosa*. The ${\mathrm{CFR}}_{\mathrm{CA}}^{50\%}$ (50%*f*T > MIC/50%*f*T > C_T_) of the CA standard regimen was 90.31% for *Escherichia coli*; 2000 mg/500 mg/8 h by 4 h infusion and 2500 mg/625 mg/8 h by 3 h infusion can achieve a ${\mathrm{CFRs}}_{\mathrm{CA}}^{100\%}$ (100%*f*T > MIC/50%*f*T > C_T_) of 90.90% and 90.96%, respectively (Table 1). The ${\mathrm{CFRs}}_{\mathrm{CA}}^{50\%}$ and ${\mathrm{CFRs}}_{\mathrm{CA}}^{100\%}$ of all simulated regimens were <90% for *Klebsiella pneumoniae*. For *Pseudomonas aeruginosa*, the regimen of 2500 mg/625 mg/6 h by 2 h infusion and 3000 mg/750 mg/8 h by 3 h infusion achieved a ${\mathrm{CFRs}}_{\mathrm{CA}}^{50\%}$ of 90.76% and 90.91%, respectively; Furthermore, the regimen of 3500 mg/875 mg/6 h by 4 h infusion and 4000 mg/1000 mg/6 h by 2 h infusion achieved a ${\mathrm{CFRs}}_{\mathrm{CA}}^{100\%}$ of 90.11% and 90.34%, respectively.

For the infections caused by *Escherichia coli*, the ${\mathrm{CFR}}_{\mathrm{CT}}^{40\%}$ (40%*f*T > MIC/20%*f*T > C_T_) and ${\mathrm{CFR}}_{\mathrm{CT}}^{100\%}$ (100%*f*T > MIC/20%*f*T > C_T_) of the CT standard regimen were 98.72% and 92.21%, respectively (Table 2). For *Klebsiella pneumoniae* and *Pseudomonas aeruginosa*, the ${\mathrm{CFRs}}_{\mathrm{CT}}^{40\%}$ and ${\mathrm{CFRs}}_{\mathrm{CT}}^{100\%}$ of all simulated regimens were <90%. 

The ${\mathrm{CFR}}_{\mathrm{MV}}^{45\%}$ (45%*f*T > MIC/*f*AUC/MIC > 9) of the MV standard regimen for *Escherichia coli*, *Klebsiella pneumoniae* and *Pseudomonas aeruginosa* was 97.48%, 96.44% and 94.04%, respectively (Table 3). The ${\mathrm{CFR}}_{\mathrm{MV}}^{100\%}$ (100%*f*T > MIC/*f*AUC/MIC > 9) of the standard regimen for *Escherichia coli* and *Klebsiella pneumoniae* was 95.37% and 91.98%, respectively. However, for *Pseudomonas aeruginosa*, the regimen of 2000 mg/2000 mg/6 h by 5 h infusion, 2500 mg /2500 mg/6 h by 4 h infusion, 3000 mg/3000 mg/6 h by 3 h infusion and 4000 mg/4000 mg/8 h by 5 h infusion afforded ${\mathrm{CFRs}}_{\mathrm{MV}}^{100\%}$ estimates of 91.71%, 91.86%, 90.86% and 90.59%, respectively.

## 3. Discussion

Although the application of novel antibacterial agents is promising, studies have reported that certain GNB showed resistance to novel BLBLIs [9,10,11]. The distribution of MIC presented in this study also indicated that a fraction of *Escherichia coli*, *Klebsiella pneumoniae,* and *Pseudomonas aeruginosa* strains were resistant to novel BLBLIs (Table 4). Additionally, the PK of novel BLBLIs in critically ill patients can be significantly altered. Therefore, it is essential to optimize their regimens in critically ill patients.

In the present study, we found that the CA standard regimen (2000 mg/500 mg/8 h, 2 h infusion) achieved optimal ${\mathrm{PTA}}_{\mathrm{CA}}^{50\%}$ at MIC ≤ 8 mg/L (Figure 1), which was consistent with the clinical breakpoint of CA for *Enterobacteriaceae* and *Pseudomonas aeruginosa* [15]. A prospective clinical study in patients with severe cystic fibrosis infection showed that when PK/PD targets of ceftazidime and avibactam were 50%*f*T > MIC and 50%*f*T > C_T_ (1 mg/L), the standard regimen achieved a >90% joint PTA at a MIC of 16 mg/L, which was similar to the results of this study (${\mathrm{PTA}}_{\mathrm{CA}}^{50\%}$ at a MIC of 16 mg/L was 86.12% in this study) [13]. However, it is recommended to use 100%*f*T > MIC as the PK/PD target to ensure clinical efficacy in critically ill patients. Under this target, the standard regimen can ensure the ${\mathrm{PTAs}}_{\mathrm{CA}}^{100\%}$ > 90% when MIC ≤ 4 mg/L, and it is necessary to adjust the regimen to achieve optimal ${\mathrm{CFRs}}_{\mathrm{CA}}^{100\%}$ regardless of the type of infected microorganisms. A study evaluated the clinical activity of CA in patients caused by MDR pathogens, which found that the favorable microbiological response of patients caused by *Escherichia coli*, *Klebsiella pneumoniae,* and *Pseudomonas aeruginosa* were 79.3%, 78.9%, and 59.1%, and the susceptibility (isolates with MICs ≤ 8 mg/L) rate of the above-mentioned strains for CA were 100%, 98.4%, and 66.1%, respectively [16]. However, the MIC data used in this study indicated that the resistance rates of *Escherichia coli*, *Klebsiella pneumoniae* and *Pseudomonas aeruginosa* for CA were 8.38%, 13.57%, and 19.44%, respectively (Table 4), and the ${\mathrm{CFRs}}_{\mathrm{CA}}^{100\%}$ of standard regimens were suboptimal for the three strains. The higher resistance rate of *Klebsiella pneumoniae* and *Pseudomonas aeruginosa* may be due to the diversity of drug-resistance mechanisms, such as the production of β-lactamase that avibactam does not inhibit, loss of porin, and overexpression of efflux pump [17,18]. Furthermore, the ${\mathrm{CFRs}}_{\mathrm{CA}}^{100\%}$ of all simulated regimens were < 90% for *Klebsiella pneumoniae* for a number of strains (12.9%) with a MIC of 128 mg/L. The results of ${\mathrm{PTAs}}_{\mathrm{CA}}^{100\%}$ or ${\mathrm{CFRs}}_{\mathrm{CA}}^{100\%}$ < 90% indicated that increased dosages, change in other antibacterial agents, or combined therapy are needed for patients with infection caused by the MDR strains. The clinical breakpoints of CT for *Enterobacteriaceae* and *Pseudomonas aeruginosa* were 2 mg/L and 4 mg/L [19]. Our study found that the CT standard regimen achieved optimal ${\mathrm{PTAs}}_{\mathrm{CT}}^{40\%}$ at a MIC ≤ 8 mg/L (Figure 1), which was consistent with a PK study of Japanese patients (the PK/PD target of ceftolozane was 30%*f*T > MIC) [20]. A pooled study of CT therapy in patients caused by extended-spectrum β-Lactamases (ESBL)-producing *Escherichia coli* and *Klebsiella pneumoniae* found that the microbiological eradications of *Escherichia coli* and *Klebsiella pneumoniae* were 82% and 77.8%, respectively. However, the MIC distribution data used in this study showed that the resistance rates of *Escherichia coli* and *Klebsiella pneumoniae* for CT were 3.55% and 26.84%, respectively (Table 4). The standard regimen only achieved the target ${\mathrm{CFRs}}_{\mathrm{CT}}^{100\%}$ for *Escherichia coli*, and all regimens failed to achieve the PK/PD targets for *Klebsiella pneumoniae*. The suboptimal results of *Klebsiella pneumoniae* may be due to the mechanism of resistance of this isolate, such as the production of ESBL and oxacillinase, which tazobactam could not inhibit [21]. CT is recommended used for infection caused by *Pseudomonas aeruginosa*. However, in this study, the resistance rate of *Pseudomonas aeruginosa* for CT was 17.93%, and the CFRs were all suboptimal (Table 2 and Table 4), which was related to the resistance mechanism that producing a chromosomally encoded class C cephalosporinase often responsible for the resistance to β-lactam antibiotics [22]. The PTAs results suggested that CT should be selected for treatment according to MIC values. Meanwhile, in the presence of high-risk factors of *Klebsiella pneumoniae* and *Pseudomonas aeruginosa* resistance, CT should be carefully selected for empirical treatment.

The MV standard regimen achieved optimal ${\mathrm{PTA}}_{\mathrm{MV}}^{45\%}$ at the clinical breakpoint of MV for GNB (4 mg/L) (Figure 1). Under different PK/PD targets, the MV standard regimen could achieve optimal CFRs for *Escherichia coli* and *Klebsiella pneumoniae*. Vaborbactam, a cyclic boronic acid β-lactamase inhibitor, has documented activity in combination with meropenem against KPC-producing *Enterobacteriaceae* [23]. However, the results of *Pseudomonas aeruginosa* were suboptimal, which may be due to the resistance mechanisms that were not antagonized by vaborbactam [23]. 

According to the results of MCS, the CFR results of three antibacterial agents were similar for *Escherichia coli*, but MV was superior for *Klebsiella pneumoniae* and *Pseudomonas aeruginosa*. Our previous Meta-analysis also showed that the comprehensive effectiveness of MV, CA, and CT were excellent, and MV was better than CA and CT, but there was no significant difference [24]. Notably, the three novel BLBLIs have a different spectrum of activity and specific indications. CA is used for *Enterobacteriaceae* that produced class A carbapenemase and some of the class D carbapenemase and *Pseudomonas aeruginosa*; CT for *Pseudomonas aeruginosa*; MV for class A carbapenemase [25]. These new BLBLIs will not work on some resistance mechanisms such as class B carbapenemase, regardless of the PK characteristics. Therefore, BLBLIs should be selected according to specific pathogenic bacteria and drug resistance mechanisms. 

Our study also had limitations. (1) Currently, PK studies of MV in critically ill patients have not been reported. The PK data of MV used in MCS was from adult patients. However, the PK parameters used in the present study for meropenem were similar to those from a study in critically ill patients with severe sepsis and septic shock [26]. Vaborbactam is mainly excreted by the kidneys. The PTA of the standard regimen of vaborbactam was far more than 90%. However, many critically ill patients may have renal failure, which will increase the PK/PD target of vaborbactam. Therefore, our results can still be applied to critically ill patients. (2) Although MCS is a useful tool for determining appropriate empirical antibiotic dosage regimens at nationally, regional levels, further clinical trials are needed to validate the efficacy and safety of higher dosages and extended infusions.

## 4. Materials and Methods

### 4.1. PK Parameters

PK parameters for CA and CT in critically ill patients were derived from a single-center phase IV clinical study in the USA and an observational study in Australia, respectively [27,28]. No study was retrieved on the PK parameters of MV in critically ill patients, and a study of adult patients was analyzed [29,30]. Data collected included clearance, volume of distribution, and AUC_24_ and was expressed as mean ± standard deviation.

### 4.2. PD Data 

*Enterobacteriaceae* (*Escherichia coli*, *Klebsiella pneumoniae*) and *Pseudomonas aeruginosa* are the most common pathogens of GNB infections. The MIC distribution of CA and CT for the above-mentioned strains was obtained from Eucast (https://mic.eucast.org, extracted on 5 March 2021) (Table 4). The MIC of MV for *Enterobacteriaceae* and *Pseudomonas aeruginosa* was obtained from a study published in 2017 by Castanheira et al. [31], which analyzed the activity of MV for GNB globally using a micro-broth dilution assay.

### 4.3. PK/PD Targets

The PK/PD targets for ceftazidime, ceftolozane and meropenem were 50%*f*T > MIC, 40%*f*T > MIC, and 45%*f*T > MIC, respectively [13,32,33]. The PK/PD targets of the three antibacterial agents should be increased to 70%*f*T > MIC and 100%*f*T > MIC in critically ill patients [34]. The result of an in vivo mouse infection model and an in vitro hollow-fiber model indicated that the C_T_ of avibactam was 1 mg/L, and the recommended PK/PD target was 50%*f*T > C_T_ [20,21]. Meanwhile, the C_T_ of tazobactam was 1 mg/L and the PK/PD target was 20%*f*T > C_T_ [20]. The PK/PD target of vaborbactam was *f*AUC_24_/MIC > 9 [35]. The free drug fractions used in these simulations were 85%, 92%, 98%, and 67% for ceftazidime, avibactam, meropenem, and vaborbactam, respectively [33,36]. Notably, the original PK parameter of ceftolozane and tazobactam were obtained from unbound concentrations, and their PK data were used directly for analysis in this study [28].

### 4.4. Monte Carlo Simulation

In the present study, MCS with 10000 replicates were performed using Crystal Ball software (Oracle Corporation, version 11.1.2.4, Redwood Shores, CA, USA) and the PK parameters were defined as log-normal distributions [37]. According to different PK/PD targets, the results of CA were expressed as ${\mathrm{PTA}}_{\mathrm{CA}}^{50\%}$ and ${\mathrm{CFR}}_{\mathrm{CA}}^{50\%}$ (50%*f*T > MIC/50%*f*T > C_T_), ${\mathrm{PTA}}_{\mathrm{CA}}^{70\%}$ and ${\mathrm{CFR}}_{\mathrm{CA}}^{70\%}$ (70%*f*T > MIC/50%*f*T > C_T_), and ${\mathrm{PTA}}_{\mathrm{CA}}^{100\%}$ and ${\mathrm{CFR}}_{\mathrm{CA}}^{100\%}$ (100%*f*T > MIC/50%*f*T > C_T_). The results of CT were expressed as ${\mathrm{PTA}}_{\mathrm{CT}}^{40\%}$ and ${\mathrm{CFR}}_{\mathrm{CT}}^{40\%}$ (40%*f*T > MIC/20%*f*T > C_T_), ${\mathrm{PTA}}_{\mathrm{CT}}^{70\%}$ and ${\mathrm{CFR}}_{\mathrm{CT}}^{70\%}$ (70%*f*T > MIC/20%*f*T > C_T_), and ${\mathrm{PTA}}_{\mathrm{CT}}^{100\%}$ and ${\mathrm{CFR}}_{\mathrm{CA}}^{100\%}$ (100%*f*T > MIC/20%*f*T > C_T_). The results of MV were ${\mathrm{PTA}}_{\mathrm{MV}}^{45\%}$ and ${\mathrm{CFR}}_{\mathrm{MV}}^{45\%}$ (45%*f*T > MIC/*f*AUC/MIC > 9), ${\mathrm{PTA}}_{\mathrm{MV}}^{70\%}$ and ${\mathrm{CFR}}_{\mathrm{MV}}^{70\%}$ (70%*f*T > MIC/*f*AUC/MIC > 9), and ${\mathrm{PTA}}_{\mathrm{MV}}^{100\%}$ and ${\mathrm{CFR}}_{\mathrm{MV}}^{100\%}$ (100%*f*T > MIC/*f*AUC/MIC > 9). The regimens corresponding to the joint PTA and CFR ≥ 90% were considered appropriate [13].

## 5. Conclusions

In conclusion, the standard regimens of CA, CT, and MV achieve optimal ${\mathrm{PTA}}_{\mathrm{CA}}^{50\%}$, ${\mathrm{PTA}}_{\mathrm{CT}}^{40\%}$ and ${\mathrm{PTA}}_{\mathrm{MV}}^{45\%}$ at the clinical breakpoint for *Enterobacteriaceae* and *Pseudomonas aeruginosa*. However, ${\mathrm{PTA}}_{\mathrm{CA}}^{100\%}$, ${\mathrm{PTA}}_{\mathrm{CT}}^{100\%}$ and ${\mathrm{PTA}}_{\mathrm{MV}}^{100\%}$ at the clinical breakpoint for *Enterobacteriaceae* and *Pseudomonas aeruginosa* were suboptimal. The standard regimens of CA and CT can achieve the target of ${\mathrm{CFR}}_{\mathrm{CA}}^{50\%}$, ${\mathrm{CFR}}_{\mathrm{CT}}^{40\%}$, and ${\mathrm{CFR}}_{\mathrm{CT}}^{100\%}$ for *Escherichia coli*. However, ${\mathrm{CFR}}_{\mathrm{CA}}^{100\%}$ was achieved when extending the infusion time; for *Klebsiella pneumoniae* and *Pseudomonas aeruginosa*, all simulated regimens cannot achieve optimal CFRs. The MV standard regimen achieved was optimal for *Enterobacteriaceae*, and the ${\mathrm{CFR}}_{\mathrm{MV}}^{100\%}$ of *Pseudomonas aeruginosa* was optimal when increasing the frequency and dosage of administration and prolonging the infusion time. Further large-scale and high-quality clinical studies should be conducted to validate the efficacy and safety of higher dosages and extended infusions for critically ill patients caused by GNB.

## Figures and Tables

**Figure 1 antibiotics-10-00993-f001:**
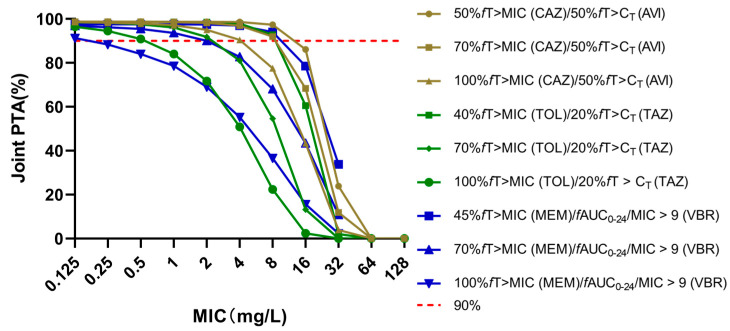
The joint PTA of different PK/PD indexes was achieved at a specific MIC for three novel BLBLIs standard regimens. Joint PTA: the product of the probability of target attainments of β-lactam and β-lactamase inhibitor; *f*T > MIC: the percentage of time during the dosing interval that free drug concentrations remain above minimal inhibitory concentrations; *f*T > C_T_: the percentage of time during the dosing interval that free drug concentrations remain above threshold concentration; *f*AUC_24_/MIC: the 24-h area under the free concentration-time curve divided by the minimum inhibitory concentration ratio; CAZ: ceftazidime; AVI: avibactam; TOL: ceftolozane; TAZ: tazobactam; MEM: meropenem: VBR: vaborbactam.

**Table 1 antibiotics-10-00993-t001:** Cumulative fraction of responses for ceftazidime/avibactam for *Enterobacteriaceae* and *Pseudomonas aeruginosa*.

Dose (mg)	${\mathrm{CFR}}_{\mathrm{CA}}^{50\%}$				${\mathrm{CFR}}_{\mathrm{CA}}^{70\%}$				${\mathrm{CFR}}_{\mathrm{CA}}^{100\%}$			
	2 h	3 h	4 h	Continuous Infusion	2 h	3 h	4 h	Continuous Infusion	2 h	3 h	4 h	Continuous Infusion
*Escherichia coli*												
2000/500 q8h	**90.31%**	92.65%	93.07%	93.11%	88.65%	**92.05%**	92.64%	92.93%	84.54%	89.94%	**90.90%**	92.35%
2500/625 q8h	91.51%	93.08%	93.25%	93.25%	**90.11%**	92.68%	92.97%	93.16%	86.42%	**90.96%**	91.64%	92.78%
*Klebsiella pneumoniae*												
2000/500 q8h	84.58%	86.60%	87.00%	87.06%	82.53%	86.04%	86.64%	86.96%	77.63%	83.71%	84.73%	86.45%
2000/500 q6h	86.95%	87.09%	87.12%	87.10%	86.73%	86.95%	87.03%	87.05%	85.72%	86.19%	86.48%	86.85%
2500/625 q8h	85.70%	86.95%	87.12%	87.12%	84.10%	86.60%	86.89%	87.08%	79.79%	84.74%	85.51%	86.80%
2500/625 q6h	87.06%	87.12%	87.14%	87.13%	86.92%	87.04%	87.11%	87.12%	86.16%	86.5%	86.81%	87.01%
3000/750 q8h	86.27%	87.05%	87.13%	87.14%	85.09%	86.78%	86.99%	87.12%	81.57%	85.26%	85.90%	87.10%
3000/750 q6h	87.09%	87.14%	87.14%	87.14%	87.01%	87.11%	87.13%	87.13%	86.43%	86.74%	86.93%	87.11%
3500/875 q8h	86.58%	87.08%	87.14%	87.14%	85.71%	86.87%	87.07%	87.14%	82.65%	85.58%	86.26%	87.04%
3500/875 q6h	87.13%	87.15%	87.15%	87.16%	87.06%	87.12%	87.14%	87.15%	86.55%	86.87%	87.00%	87.12%
4000/1000 q8h	86.74%	87.12%	87.16%	87.14%	86.01%	86.95%	87.07%	87.14%	83.33%	85.90%	86.31%	87.09%
4000/1000 q6h	87.15%	87.17%	87.17%	87.15%	87.09%	87.12%	87.14%	87.14%	86.71%	86.88%	87.01%	87.13%
*Pseudomonas aeruginosa*												
2000/500 q8h	80.28%	87.4%	87.73%	87.60%	77.07%	85.13%	85.89%	86.49%	70.58%	80.42%	81.73%	84.23%
2000/500 q6h	88.86%	88.86%	88.82%	88.62%	87.20%	87.48%	87.64%	87.63%	84.16%	84.88%	85.21%	86.07%
2500/625 q8h	82.80%	89.47%	89.64%	89.47%	79.77%	87.35%	87.71%	88.40%	73.52%	82.90%	83.92%	86.23%
2500/625 q6h	**90.76%**	90.73%	90.65%	90.41%	89.13%	89.44%	89.47%	89.50%	86.25%	86.8%	87.3%	87.85%
3000/750 q8h	84.73%	**90.91%**	91.08%	90.97%	81.90%	88.90%	89.29%	89.94%	76.23%	84.73%	85.77%	88.86%
3000/750 q6h	92.09%	92.05%	91.99%	91.69%	**90.67%**	90.82%	90.89%	90.84%	87.94%	88.4%	88.92%	89.24%
3500/875 q8h	85.59%	92.05%	92.22%	92.05%	83.39%	**90.08%**	90.56%	91.09%	78.22%	86.15%	87.26%	89.16%
3500/875 q6h	93.19%	93.10%	93.08%	92.84%	91.78%	91.91%	92.06%	91.99%	89.10%	89.61%	**90.11%**	90.56%
4000/1000 q8h	87.43%	93.04%	93.21%	93.03%	84.96%	91.11%	91.47%	92.10%	79.84%	87.44%	88.23%	90.12%
4000/1000 q6h	94.24%	94.11%	93.96%	93.70%	92.79%	92.89%	92.93%	92.98%	**90.34%**	90.65%	91.00%	91.50%

${\mathrm{CFR}}_{\mathrm{CA}}^{50\%}$: 50%*f*T > MIC/50%*f*T > C_T_; ${\mathrm{CFR}}_{\mathrm{CA}}^{70\%}$: 70%*f*T > MIC/50%*f*T > C_T_; ${\mathrm{CFR}}_{\mathrm{CA}}^{100\%}$: 100%*f*T > MIC/50%*f*T > C_T_. Grey: PTA values ≥ 90%; Standard regimen of ceftazidime/avibactam: 2000 mg/500 mg/8 h, 2 h infusion.

**Table 2 antibiotics-10-00993-t002:** Cumulative fraction of responses for ceftolozane/tazobactam for *Enterobacteriaceae* and *Pseudomonas aeruginosa*.

Dose (mg)	${\mathrm{CFR}}_{\mathrm{CT}}^{40\%}$				${\mathrm{CFR}}_{\mathrm{CT}}^{70\%}$				${\mathrm{CFR}}_{\mathrm{CT}}^{100\%}$			
	1 h	3 h	4 h	Continuous Infusion	1 h	3 h	4 h	Continuous Infusion	1 h	3 h	4 h	Continuous Infusion
*Escherichia coli*												
1000/500 q8h	**98.72%**	98.99%	99.02%	99.00%	**97.18%**	98.21%	98.47%	98.78%	**92.21%**	95.11%	96.49%	98.35%
1250/625 q8h	98.93%	99.16%	99.17%	99.16%	97.61%	98.54%	98.69%	98.95%	93.13%	96.10%	97.02%	98.60%
*Klebsiella pneumoniae*												
1000/500 q8h	80.78%	81.28%	81.38%	81.26%	76.76%	78.40%	78.99%	80.04%	70.09%	73.37%	75.12%	78.45%
1000/500 q6h	82.06%	82.11%	81.98%	81.66%	79.39%	80.25%	80.45%	80.53%	75.91%	77.86%	78.47%	79.22%
1250/625 q8h	81.97%	82.48%	82.50%	82.44%	77.87%	79.50%	79.95%	81.01%	71.63%	75.00%	76.17%	79.35%
1250/625 q6h	83.59%	83.44%	83.25%	83.12%	80.46%	81.25%	81.37%	81.79%	77.10%	78.91%	79.44%	80.32%
1500/750 q8h	83.27%	83.67%	83.75%	83.62%	78.83%	80.30%	80.77%	81.95%	73.04%	75.90%	77.09%	80.12%
1500/750 q6h	85.03%	85.04%	84.82%	84.33%	81.45%	82.21%	82.51%	82.70%	78.17%	79.75%	80.31%	81.12%
1750/875 q8h	84.47%	85.02%	84.98%	84.84%	79.71%	81.16%	81.60%	82.93%	73.97%	76.87%	77.84%	80.82%
1750/875 q6h	86.30%	86.35%	86.27%	85.83%	82.31%	83.28%	83.61%	83.89%	78.84%	80.48%	81.13%	81.98%
2000/1000 q8h	85.55%	85.83%	86.26%	86.00%	80.51%	81.87%	82.41%	83.75%	74.96%	77.66%	78.65%	81.50%
2000/1000 q6h	87.7%	87.53%	87.26%	86.96%	83.40%	84.31%	84.39%	84.87%	79.69%	81.28%	81.72%	82.70%
*Pseudomonas aeruginosa*												
1000/500 q8h	86.19%	86.73%	86.79%	86.74%	80.56%	83.64%	84.82%	86.16%	68.29%	74.13%	77.55%	84.42%
1000/500 q6h	83.02%	80.57%	77.64%	71.51%	80.84%	79.62%	77.02%	71.15%	75.39%	76.51%	74.90%	70.35%
1250/625 q8h	86.72%	87.16%	87.17%	87.15%	82.05%	84.92%	85.64%	86.63%	71.35%	77.11%	79.27%	85.46%
1250/625 q6h	86.62%	86.10%	83.44%	82.80%	84.79%	85.26%	82.82%	82.40%	79.91%	82.68%	81.22%	81.72%
1500/750 q8h	87.18%	87.50%	87.54%	87.50%	83.17%	85.52%	86.18%	86.98%	73.53%	78.61%	80.76%	86.03%
1500/750 q6h	87.67%	87.73%	86.85%	85.6%	86.02%	86.83%	86.18%	85.14%	81.90%	84.69%	84.8%	84.55%
1750/875 q8h	87.65%	87.91%	87.87%	87.83%	83.95%	86.06%	86.56%	87.29%	75.08%	79.96%	81.74%	86.44%
1750/875 q6h	88.32%	88.26%	88.12%	88.16%	86.48%	87.28%	87.31%	87.07%	82.62%	85.29%	86.08%	86.42%
2000/1000 q8h	88.14%	88.10%	88.37%	88.25%	84.67%	86.45%	86.92%	87.53%	76.45%	81.03%	82.76%	86.75%
2000/1000 q6h	89.24%	88.95%	88.81%	88.43%	87.02%	87.67%	87.73%	87.58%	83.54%	85.87%	86.55%	86.92%

${\mathrm{CFR}}_{\mathrm{CT}}^{40\%}$: 40%*f*T > MIC/20%*f*T > C_T_; ${\mathrm{CFR}}_{\mathrm{CT}}^{70\%}$: 70%*f*T > MIC/20%*f*T > C_T_; ${\mathrm{CFR}}_{\mathrm{CA}}^{100\%}$: 100%*f*T > MIC/20%*f*T > C_T_.Grey: PTA values ≥ 90%; Standard regimen of ceftolozane/tazobactam: 1000 mg/500 mg/8 h, 1 h infusion.

**Table 3 antibiotics-10-00993-t003:** Cumulative fraction of responses for meropenem/vaborbactam for *Enterobacteriaceae* and *Pseudomonas aeruginosa*

Dose (mg)	${\mathrm{CFR}}_{\mathrm{MV}}^{45\%}$			${\mathrm{CFR}}_{\mathrm{MV}}^{70\%}$			${\mathrm{CFR}}_{\mathrm{MV}}^{100\%}$		
	3 h	4 h	5 h	3 h	4 h	5 h	3 h	4 h	5 h
*Escherichia coli*									
2000/2000 q8h	**97.48%**	97.49%	97.49%	**97.35%**	97.47%	97.48%	**95.37%**	96.52%	97.10%
2500/2500 q8h	99.24%	99.24%	99.24%	99.14%	99.23%	99.23%	97.64%	98.23%	98.95%
*Klebsiella pneumoniae*									
2000/2000 q8h	**96.44%**	96.60%	96.65%	**95.49%**	95.86%	96.06%	**91.98%**	93.78%	94.70%
2500/2500 q8h	98.52%	98.64%	98.78%	97.54%	97.82%	98.07%	94.66%	95.61%	96.83%
*Pseudomonas aeruginosa*									
2000/2000 q8h	**94.04%**	94.98%	95.30%	87.62%	**90.16%**	91.92%	75.11%	79.50%	82.86%
2000/2000 q6h	95.87%	95.84%	96.08%	**92.64%**	93.62%	94.76%	86.40%	88.47%	**91.71%**
2500/2500 q8h	97.04%	97.65%	98.09%	**91.18%**	93.12%	95.03%	79.63%	82.70%	86.67%
2500/2500 q6h	98.38%	98.61%	98.52%	95.55%	96.76%	97.56%	89.14%	**91.86%**	94.88%
3000/3000 q8h	98.10%	98.68%	99.02%	92.73%	94.72%	96.24%	81.49%	84.78%	88.23%
3000/3000 q6h	99.24%	99.43%	99.45%	96.95%	97.96%	98.84%	**90.86%**	93.54%	96.58%
3500/3500 q8h	98.60%	99.20%	99.50%	93.31%	95.61%	97.28%	82.30%	86.11%	89.69%
3500/3500 q6h	99.66%	99.78%	99.83%	97.73%	98.67%	99.44%	92.37%	94.51%	97.48%
4000/4000 q8h	99.16%	99.53%	99.72%	94.72%	96.26%	97.78%	84.76%	86.98%	**90.59%**
4000/4000 q6h	99.80%	99.87%	99.92%	98.19%	98.96%	99.66%	93.03%	95.14%	98.18%

${\mathrm{CFR}}_{\mathrm{MV}}^{45\%}$: 45%*f*T > MIC/*f*AUC/MIC > 9; ${\mathrm{CFR}}_{\mathrm{MV}}^{70\%}$: 70%*f*T > MIC/*f*AUC/MIC > 9; ${\mathrm{CFR}}_{\mathrm{MV}}^{100\%}$: 100%*f*T > MIC/*f*AUC/MIC > 9. Grey: PTA values ≥ 90%; Standard regimen of meropenem/vaborbactam: 2000 mg/2000 mg/8 h, 3 h infusion.

**Table 4 antibiotics-10-00993-t004:** The MIC distribution of ceftazidime/avibactam, ceftolozane/tazobactam and meropenem/vaborbactam for *Enterobacteriaceae* and *Pseudomonas aeruginosa.*

MIC	≤0.015	0.03	0.06	0.125	0.25	0.5	1	2	4	8	16	32	64	128
ceftazidime/avibactam														
*E. coli*	-	-	10	29	25	23	23	32	16	6	3	0	2	10
*KP*	-	-	3	8	17	16	31	26	11	9	1	0	0	18
*PA*	-	-	0	0	1	20	96	82	42	20	22	17	16	8
ceftolozane/tazobactam														
*E. coli*	-	1	80	1995	1820	691	343	119	70	45	27	27	17	-
*KP*	-	-	28	576	866	447	292	175	116	80	98	175	400	6 ^a^
*PA*	-	-	-	8	150	1738	1528	737	533	225	68	82	645	10 ^a^
meropenem/vaborbactam														
*E. coli*	3916	274	32	2	4	3	4	1	1	0	0	1 ^b^		
*KP*	819	928	32	41	50	47	28	9	12	6	7	31 ^b^		
*PA*	41	78	245	442	424	438	234	159	185	138	125	95 ^b^		

*E. coli: Escherichia coli; KP: Klebsiella pneumoniae; PA: Pseudomonas aeruginosa*. ^a^ MIC ≥ 128; ^b^ MIC ≥ 32.

## Data Availability

All data are applicable in the paper.

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
