# Peer review of "Pharmacokinetic/Pharmacodynamic Adequacy of Novel β-Lactam/β-Lactamase Inhibitors against Gram-Negative Bacterial in Critically Ill Patients"

_antibiotics, 2021, doi:10.3390/antibiotics10080993_

Round 1
Reviewer 1 Report
Han et al. compared different regimens of novel β-lactam/β-lactamase inhibitors using Monte Carlo simulations. The manuscript is well written and of interest for physicians caring for critically ill patients. I only have minor points to suggest for the discussion.
Minor points
- Ceftazidime/avibactam (CA), ceftolozane/tazobactam (CT), and meropenem/vaborbactam (MV) have different spectrum of activity and specific indications. CA is used for class A carbapenemase (e.g., KPC), class D carbapenemase (e.g., OXA-48) and MDR aeruginosa; CT for MDR P. aeruginosa; MV for class A carbapenemase and MDR P. aeruginosa (Doi Y CID 2019, PMID: 31724043). This could be added to the discussion, because it is expected that these new antibiotics won’t work on some resistance mechanisms such as class B carbapenemase (e.g., NDM), regardless of the PK characteristics.
- Recommended standard dosage / high dosage regimens could be highlighted in Tables 1, 2, and 3, or this could be added to the Table legends to facilitate reading.
- For MV, no study on PK parameters was available for critically ill patients. However, identifying optimal regimens in critically ill patients is an objective of the study (Line 57). This should be discussed as a limit of this study.
Author Response
Point 1: Ceftazidime/avibactam (CA), ceftolozane/tazobactam (CT), and meropenem/vaborbactam (MV) have different spectrum of activity and specific indications. CA is used for class A carbapenemase (e.g., KPC), class D carbapenemase (e.g., OXA-48) and MDR aeruginosa; CT for MDR P. aeruginosa; MV for class A carbapenemase and MDR P. aeruginosa (Doi Y CID 2019, PMID: 31724043). This could be added to the discussion, because it is expected that these new antibiotics won’t work on some resistance mechanisms such as class B carbapenemase (e.g., NDM), regardless of the PK characteristics.
Response 1: We agree with the Reviewer’s comment. We have added the following explanation in the Discussion in our revised manuscript (Page 8, Lines 186-192).
Notably, the three novel BLBLIs have different spectrum of activity and specific indications. CA is used for Enterobacteriaceae that producted class A carbapenemase and some of the class D carbapenemase and Pseudomonas aeruginosa; CT for Pseudomonas aeruginosa; MV for class A carbapenemase. These new BLBLIs won’t work on some resistance mechanisms such as class B carbapenemase, regardless of the PK characteristics. Therefore, BLBLIs should be selected according to specific pathogenic bacteria and drug resistance mechanism.
Point 2: Recommended standard dosage / high dosage regimens could be highlighted in Tables 1, 2, and 3, or this could be added to the Table legends to facilitate reading.
Response 2: Thanks. We have added the following explanation in the introduction in our revised manuscript:
Standard regimen of ceftazidime/avibactam: 2000 mg/500 mg/8 h, 2 h infusion. (Page 4, Line 123).
Standard regimen of ceftolozane/tazobactam: 1000 mg/500 mg/8 h, 1 h infusion. (Page 5, Line 127).
Standard regimen of meropenem/vaborbactam: 2000 mg/2000 mg/8 h, 3 h infusion. (Page 6, Line 132).
Point 3: For MV, no study on PK parameters was available for critically ill patients. However, identifying optimal regimens in critically ill patients is an objective of the study (Line 57). This should be discussed as a limit of this study.
Response 3: Thanks. We have added the following explanation in the introduction in our revised manuscript (Page 8, Lines 205-211).
Currently, PK studies of MV in critically ill patients have not been reported. The PK data of MV used in MCS was from adult patients. However, the PK parameters used in the present study for meropenem were similar to those from a study in critically ill patients with severe sepsis and septic shock. Vaborbactam is mainly excreted by the kidneys. The PTA of the standard regimen of vaborbactam was far more than 90%, however, many critically ill patients may have renal failure, which will increase PK/PD target of vaborbactam. Therefore, our results can still be applied to critically ill patients.
Reviewer 2 Report
The authors state that recommended dosage regimen of antibiotics is derived from PK studies in healthy individuals. In this manuscript Han et al. make the argument critically ill patients undergo physiological/pathological changes which change PK parameters of drugs. As such the authors wanted to evaluate the ability of current antibiotic regimens to reach PK/PD targets. The authors obtained published PK/PD data of different drug regimens and performed simulation studies to determine two main parameters for the drug regimens: probability of target attainment (PTA) and cumulative fraction of response (CFR). This analysis was specifically focused on combined beta lactams and beta lactamase inhibitor drug regimens. In general, the results of this study is straightforward and main conclusion is that the current standard regimens of CA, CT and MV only achieve 50%, 40% and 45% PTA, respectively, and is not efficient to reach 100% PTA. The main drawback of this manuscript is that many of the terms and conditions are not explained very clearly. The results section is very difficult to follow and easily loses interest of a reader who is not directly involved in drug discovery research. It is recommended to explain PTA, CFR, PK/PD index of the percentage of time that free drug concentration is above the MIC (fT>MIC) and relevance of these parameters in the introduction. This will set the stage for the reader and make it easier to understand the results of this study. It is recommended to accept the manuscript for publication after these minor revisions.
Author Response
Point : The authors state that recommended dosage regimen of antibiotics is derived from PK studies in healthy individuals. In this manuscript Han et al. make the argument critically ill patients undergo physiological/pathological changes which change PK parameters of drugs. As such the authors wanted to evaluate the ability of current antibiotic regimens to reach PK/PD targets. The authors obtained published PK/PD data of different drug regimens and performed simulation studies to determine two main parameters for the drug regimens: probability of target attainment (PTA) and cumulative fraction of response (CFR). This analysis was specifically focused on combined beta lactams and beta lactamase inhibitor drug regimens. In general, the results of this study is straightforward and main conclusion is that the current standard regimens of CA, CT and MV only achieve 50%, 40% and 45% PTA, respectively, and is not efficient to reach 100% PTA. The main drawback of this manuscript is that many of the terms and conditions are not explained very clearly. The results section is very difficult to follow and easily loses interest of a reader who is not directly involved in drug discovery research. It is recommended to explain PTA, CFR, PK/PD index of the percentage of time that free drug concentration is above the MIC (fT>MIC) and relevance of these parameters in the introduction. This will set the stage for the reader and make it easier to understand the results of this study. It is recommended to accept the manuscript for publication after these minor revisions.
Response: Thanks for your comments. We have added the following explanation in the introduction in revised manuscript (Page 2, Lines 54-69):
Monte Carlo simulation (MCS) is now widely used to optimize antibacterial agents regimens by combining the pharmacokinetics and pharmacodynamics principles. The results of MCS were expressed as probability of target attainment (PTA) and cumulative fraction of response (CFR). The PTA was defined as the probability that a specific value of a PK/PD index was achieved at a certain minimal inhibitory concentrations (MIC). For BLBLIs, a joint PTA was calculated and the joint PTA was the product of the PTA of β-lactam and β-lactamase inhibitors. The CFR was defined as the expected population joint PTA for a specific drug dosage and a specific pathogen. Ceftazidime, ceftolozane and meropenem are time-dependent antibacterial agents with a PK/PD parameter of the percentage of time during the dosing interval that free drug concentrations remain above minimal inhibitory concentrations (%fT>MIC) [14]. β-lactamase inhibitors require a certain threshold concentration (CT) to exert their inhibitory effect. The PK/PD parameter of avibactam and tazobactam was the percentage of time during the dosing interval that free drug concentrations remain above threshold concentration (%fT> CT) and vaborbactam was the 24-h area under the free concentration-time curve divided by the minimum inhibitory con-centration ratio (fAUC24/MIC).
Reviewer 3 Report
This study was conducted to identify optimal regimens of novel β-lactam/β-lactamase inhibitors (BLBLIs), ceftazidime/avi-10 bactam, ceftolozane/tazobactam and meropenem/vaborbactam.
The authors explored the topic and they obtained the purpose of the study. Although large-scale, high-quality clinical trials are needed to validate the efficacy and safety of higher dosages and extended infusions
About strenghts the authors explored the topic and they obtained the purpose of the study. The paper is well written and text is clear to read.
The methods used are sufficiently documented and allow replication studies. Results obtained are well explained and data interpretation is also correct. Conclusions are consistent with the evidence and arguments presented.
I will recommend the acceptance of this manuscript in present form.
Author Response
Point: This study was conducted to identify optimal regimens of novel β-lactam/β-lactamase inhibitors (BLBLIs), ceftazidime/avibactam, ceftolozane/tazobactam and meropenem/vaborbactam.
The authors explored the topic and they obtained the purpose of the study. Although large-scale, high-quality clinical trials are needed to validate the efficacy and safety of higher dosages and extended infusions.
About strenghts the authors explored the topic and they obtained the purpose of the study. The paper is well written and text is clear to read.
The methods used are sufficiently documented and allow replication studies. Results obtained are well explained and data interpretation is also correct. Conclusions are consistent with the evidence and arguments presented.
I will recommend the acceptance of this manuscript in present form.
Response: Thanks for your comments.